# ACTIVE DEEP PROBABILISTIC SUBSAMPLING

## ABSTRACT

Subsampling a signal of interest can reduce costly data transfer, battery drain, radiation exposure and acquisition time in a wide range of problems. The recently proposed Deep Probabilistic Subsampling (DPS) method effectively integrates subsampling in an end-to-end deep learning model, but learns a static pattern for all datapoints. We generalize DPS to a sequential method that actively picks the next sample based on the information acquired so far; dubbed *Active-DPS* (A-DPS). We validate that A-DPS improves over DPS for MNIST classification at high subsampling rates. We observe that A-DPS learns to actively adapt based on the previously sampled elements, yielding different sampling sequences across the dataset. Moreover, we demonstrate strong performance in active acquisition Magnetic Resonance Image (MRI) reconstruction, outperforming DPS and other deep learning methods.

## 1 INTRODUCTION

Present-day technologies produce and consume vast amounts of data, which is typically acquired using an analog-to-digital converter (ADC). The amount of data digitized by an ADC is determined not only by the temporal sampling rate, but also by the manner in which spatial acquisitions are taken, e.g. by using a specific design of sensor arrays. Reducing the number of sample acquisitions needed, can lead to meaningful reductions in scanning time, e.g. in Magnetic Resonance Imaging (MRI), radiation exposure, e.g. in Computed Tomography (CT), battery drain, and bandwidth requirements. While the Nyquist theorem is traditionally used to provide theoretical bounds on the sampling rate, in recent years signal reconstruction from sub-Nyquist sampled data has been achieved through a framework called Compressive Sensing (CS).

First proposed by Donoho (2006), and later applied for MRI by Lustig et al. (2007), CS leverages structural signal priors, specifically sparsity under some known transform. By taking compressive measurements followed by iterative optimization of a linear system under said sparsity prior, reconstruction of the original signal is possible while sampling at sub-Nyquist rates. Researchers have employed CS with great success in a wide variety of applications, such as radar (Baraniuk & Steeghs, 2007; Ender, 2010), seismic surveying (Herrmann et al., 2012), spectroscopy (Sanders et al., 2012), and medical imaging (Han et al., 2016; Lai et al., 2016).

However, both the need to know the sparsifying basis of the data, and the iterative nature of the reconstruction algorithms, still hamper practical applicability of CS in many situations. These limitations can be overcome by the use of deep learning reconstruction models that make the sparsity assumption implicit, and facilitate non-iterative inference once trained. Moreover, the (typically random) nature of the measurement matrix in CS does, despite adhering to the given assumptions, not necessarily result in an optimal measurement given the underlying data statistics and the downstream system task. This has recently been tackled by algorithms that learn the sampling scheme from a data distribution.

In general, these data-driven sampling algorithms can be divided into two categories: algorithms that learn sampling schemes which are fixed once learned (Huijben et al., 2020a;b;c; Ravishankar & Bresler, 2011; Sanchez et al., 2020; Bahadir et al., 2019; Bahadir et al., 2020; Weiss et al., 2019), and algorithms that learn to actively sample (Ji et al., 2008; Zhang et al., 2019; Jin et al., 2019; Pineda et al., 2020; Bakker et al., 2020); selecting new samples based on sequential acquisition of the information. The former type of algorithms learn a sampling scheme that - on average - selects informative samples of all instances originating from the training distribution. However,

when this distribution is multi-modal, using one globally optimized sampling scheme, can easily be sub-optimal on instance-level. Active acquisition algorithms deal with such shifts in underlying data statistics by conditioning sampling behavior on previously acquired information from the instance (e.g. the image to be sampled). This results in a sampling sequence that varies across test instances, i.e. sampling is *adapted* to the new data. This adaptation as a result of conditioning, promises lower achievable sampling rates, or better downstream task performance for the same rate, compared to sampling schemes that operate equivalently on all data.

In this work, we extend the Deep Probabilistic Subsampling (DPS) framework (Huijben et al., 2020a) to an active acquisition framework by making the sampling procedure iterative and conditional on the samples already acquired, see Fig. 1. We refer to our method as Active Deep Probabilistic Subsampling (A-DPS). We show how A-DPS clearly exploits the ten different modalities (i.e. the digits) present in the MNIST dataset to adopts instance-adaptive sampling sequences. Moreover, we demonstrate both on MNIST (LeCun et al., 1998) and the real-world fast MRI knee dataset (Zbontar et al., 2018), that A-DPS outperforms other state-of-the-art models for learned sub-Nyquist sampling. We make all code publicly available upon publication, in order to facilitate benchmarking to all provided baselines and A-DPS in future research.

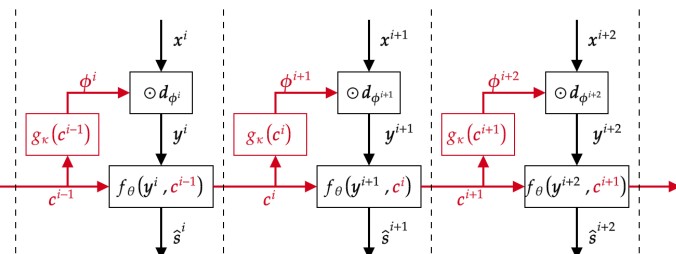

Figure 1: Architectural overview for 3 acquisition steps of A-DPS, with extensions of DPS shown in red.

## 2    RELATED WORK

Recently, several techniques for learning a fixed sampling pattern have been proposed, especially in the field of MR imaging, in which Ravishankar & Bresler (2011) were one of the firsts. In this work, the authors make use of non-overlapping cells in k-space, and move samples between these cells.During training Ravishankar & Bresler (2011) alternate between reconstruction and relocation of sampling positions. After a reconstruction step they sort the cells in terms of reconstructing error and an infinite-p norm. Selected samples from lower scoring cells are relocated to higher scoring cells in a greedy fashion.

Sanchez et al. (2020) also propose a greedy approach, in which samples are not relocated between cells, but greedily chosen to optimize a reconstruction loss on a batch of examples. Both of the types of greedy optimization do however not allow for joint learning of sampling together with a downstream reconstruction/task model, as the reconstruction has to either be parameter-free or pretrained to work well with a variety of sampling schemes.

Bahadir et al. (2019) on the other hand propose to learn the sampling pattern by thresholding pixel-based i.i.d. samples drawn from a uniform distribution, dubbed Learning-based Optimization of the Under-sampling PattErn (LOUPE). The sample rate of LOUPE is indirectly controlled by promoting sparsity through the use of an $\ell_1$ penalty on the thresholds.

One of the first active sampling schemes was proposed by Ji et al. (2008), who leverage CS reconstruction techniques that also give a measure of uncertainty of the reconstruction using Bayesian modeling. Ji et al. (2008) leveraged this uncertainty in the reconstruction to adaptivly select the next measurement that will reduce this uncertainty by the largest amount. However, this method - and other similar works from (Carson et al., 2012; Li et al., 2013) - rely on linearly combined measurements, rather than discrete sampling, with which we concern ourselves here.

In the field of MRI, Zhang et al. (2019) propose an active acquisition scheme by leveraging a reconstruction and adversarial neural network. Whereas the reconstruction network is trained to reconstruct MR images from the subsampled Fourier space (k-space), the adversarial network is trained to distinguish between already sampled, and omitted lines in this space. The k-space line that is most believed to be 'fake' (i.e. filled in by the reconstruction network) by the adversarial network, is sampled next. However, This framework only works for undersampled Fourier to image reconstruction tasks, as the discriminator requires mappings of the image in k-space. Jin et al. (2019) put forth an active acquisition scheme for MRI by leveraging reinforcement learning (RL). Two neural networks, one for sampling and one for reconstruction are trained jointly using a Monte-Carlo tree search, resulting in a sampling policy that is dependent on the current reconstruction of the image.

Concurrently with our work, both Pineda et al. (2020) and Bakker et al. (2020) proposed RL-based active acquisition techniques. Pineda et al. (2020) leverages a Double Deep Q-Network. The model is trained using a modified $\epsilon$-greedy policy, in which the best action is taken with probability $1 - \epsilon$, and an exploratory action is taken with probability $\epsilon$. Bakker et al. (2020) compare greedy with non-greedy training, finding that the greedy method leads to a higher degree of adaptability, especially for tasks with a long horizon (i.e. more samples to be taken). Both of the frameworks proposed by (Pineda et al., 2020) and Bakker et al. (2020) make use of a pretrained reconstruction network, which differs from the proposed A-DPS method that enables joint training of both the reconstruction (task) network and sampling network.

Even though subsampling is an extreme form of data compression, we differentiate from typical data compression architectures like deep encoder-decoder structures (Theis et al., 2017; Ballé et al., 2017), as these methods do not reduce data rates at the measurement stage. The feedback recurrent autoencoder proposed by Yang et al. (2020) is however related to A-DPS through its use of a recurrent context. But whereas Yang et al. (2020) learn a recurrent context to inform the encoder stage of the network, A-DPS uses this to inform the sampling pattern.

## 3 METHOD

### 3.1 GENERAL FRAMEWORK

Given a prediction task $s$ we are interested in learning to predict an optimal subsampling scheme $\boldsymbol{A} \in \{0, 1\}^{M \times N}$ (with $M \ll N$) on an input signal $\boldsymbol{x} \in \mathbb{R}^N$, resulting in a measurement $\tilde{\boldsymbol{y}} \in \mathbb{R}^M$:

$$\tilde{\boldsymbol{y}} = \boldsymbol{A}\boldsymbol{x}. \tag{1}$$

Each row in $\boldsymbol{A}$ is constrained to have $\ell_0$-norm of 1, while each column in $\boldsymbol{A}$ is constrained to have an $\ell_0$-norm of either 0 or 1, i.e. each of the $N$ candidate samples is selected at most once. In the rest of this paper we will index these candidate samples with $n \in \{1, \ldots, N\}$, and the selected samples with $m \in \{1, \ldots, M\}$. The percentage of selected samples from the candidate samples is called the sampling ratio $r = M/N \cdot 100\%$.

We also introduce a non-compressed form of the measurement $\tilde{\boldsymbol{y}}$, called $\boldsymbol{y} \in \mathbb{R}^N$, that contains $N - M$ zeros, and $M$ non-zeros at the sampled indices specified by $\boldsymbol{A}$, i.e. the *masked* input. This way, the location of samples from $\boldsymbol{x}$ is preserved, which is especially useful when $\boldsymbol{A}$ changes during training. To acquire $\boldsymbol{y}$ from $\boldsymbol{x}$, one seeks a subsampling mask $\boldsymbol{d}$ that can be applied on $\boldsymbol{x}$ via:

$$\boldsymbol{y} = \boldsymbol{d} \odot \boldsymbol{x} = \boldsymbol{A}^T \boldsymbol{A} \boldsymbol{x}, \tag{2}$$

where $\odot$ denotes an element-wise multiplication. From the resulting measurement $\boldsymbol{y}$ we then aim at predicting the downstream task $s$ through:

$$\hat{s} = f_\theta(\boldsymbol{y}), \tag{3}$$

where $f_\theta(.)$ is a function that is differentiable with respect to its input and parameters $\theta$, e.g. a neural network. Normally, optimization of the task model $f_\theta(.)$ is achieved through backpropagation of some loss function $\mathcal{L}(s, \hat{s})$. However, calculating gradients on the sampling matrix is blocked by its combinatorial nature, inhibiting joint training of the task with the sampling operation. The DPS framework provides a solution to this problem, on which we will elaborate in the next section.

## 3.2 DPS: Deep Probabilistic Subsampling

To enable joint training of the sampling operation with the downstream task model, Huijben et al. (2020a) introduce DPS. Rather than optimizing $\boldsymbol{A}$ directly, they propose to optimize a generative sampling model $P(\boldsymbol{A}|\boldsymbol{\phi})$, where $\boldsymbol{\phi}$ are learned unnormalized logits of (possibly multiple) categorical distribution(s). Each distribution expresses the probabilities for sampling any of the elements $x_n$ from $\boldsymbol{x}$ through sampling matrix $\boldsymbol{A}$. More specifically, $\phi_{m,n}$ is the log-probability for setting $a_{m,n} = 1$, and thus sampling $x_n$ as $m^{\text{th}}$ sample.

To generate a sampling pattern from these unnormalized logits, i.e. implementation of this conditional model, the Gumbel-max trick is leveraged (Gumbel, 1954). In the Gumbel-max trick the unnormalized logits are perturbed with i.i.d. Gumbel noise samples $e_{m,n} \sim \text{Gumbel}(0,1)$. By selecting the maximum of this perturbation a realization of the sampling mask can be found using:

$$\boldsymbol{A}_{m,:} = \text{one-hot}_N \left\{ \underset{n}{\text{argmax}} \left\{ w_{m-1,n} + \phi_{m,n} + e_{m,n} \right\} \right\}, \tag{4}$$

where $\boldsymbol{A}_{m,:}$ denotes the m-th row of $\boldsymbol{A}$ and one-hot$_N$ creates a one-hot vector of length $N$, with the one at the index specified by the argmax operator. Moreover, the cumulative mask $w_{m-1,n} \in \{-\infty, 0\}$ masks previously selected samples by adding minus infinity to those logits, thereby ensuring sampling without replacement.

During backpropagation, gradients are computed by relaxing this sampling procedure using the Gumbel-softmax trick (Jang et al., 2016; Maddison et al., 2017), resulting in:

$$\nabla_{\phi_m} \boldsymbol{A}_{m,:} := \nabla_{\phi_m} \mathbb{E}_{e_m} \left[ \text{softmax}_\tau \left\{ w_{m-1,n} + \phi_{m,n} + e_{m,n} \right\} \right], \tag{5}$$

where $\tau$ denotes the temperature parameter of the softmax operator. Setting $\tau > 0$ results in a smoothed sampling matrix $\boldsymbol{A}$ (i.e. elements can have values between 0 and 1 as well), allowing gradients to distribute over multiple logits during training. In the limit of $\tau \to 0$ the softmax operator approaches the one-hot argmax function of equation (4). Although this approach – also known as straight-through Gumbel-softmax – leads to biased gradients, it has been shown to work well in practice, and Huijben et al. (2020a) keep $\tau$ at a fixed value during training.

Huijben et al. (2020a) propose two regimes of DPS. First, Top-1 sampling, an expressive form of DPS where each of the $M$ selected samples are separately conditioned on all $N$ candidate samples, resulting in $M \times N$ trainable logits $\phi_{m,n}$. Second, Top-M sampling (called Top-K in their paper), a constrained form where all $M$ samples together are conditioned on all $N$ candidate samples, i.e. the logits $\phi_n$ are shared between the $M$ rows of $\boldsymbol{A}$, resulting in only $N$ trainable logits. While Top-1 sampling is more expressive, Huijben et al. (2020a) noticed slightly better results for the Top-M regime, possibly thanks to the smaller number of trainable logits, therefore facilitating optimization. For scaleability reasons, we thus choose to continue with Top-M sampling in this work and refer to this regime as DPS in the rest of this paper. We refer the reader to Huijben et al. (2020a) for more details regarding DPS.

## 3.3 A-DPS: Active Deep Probabilistic Subsampling

We have seen how DPS enables the learning of a sampling scheme that selects $M$ out of $N$ samples. However, these samples are selected simultaneously. A-DPS selects its samples in an iterative fashion, separating the logits into $I$ acquisition steps, i.e. $\boldsymbol{\phi}^i$ with $i \in \{1, 2, \dots, I\}$ and $I = M$.

Active acquisition is then achieved by introducing dependency between samples, i.e. the sampling distribution at acquisition step $i$ should depend on the information acquired in previous acquisition steps. To that end, we introduce a context vector $\boldsymbol{c_i}$, that accumulates information from all previous time steps. We then condition the sampling distribution on this context by learning a transformation $\boldsymbol{\phi} = g_\kappa(\boldsymbol{c})$, where $g_\kappa(.)$ is a function that is differentiable with respect to its input and parameters $\kappa$. Thus, instead of optimizing the parameters directly (as DPS does), we optimize $g_\kappa(\boldsymbol{c})$, which we will refer to as the sampling model.

The question then arises how to best generate this context from previous samples. Here, we follow the *analysis-by-synthesis* principle, and let the analysis (the sampling model) depend on the synthesis (the task model). This way, the task model can inform the sampling model what information it needs

to achieve its assigned task. The iterative *analysis-by-synthesis* scheme of A-DPS is formalized as follows:

$$\phi^i = g_{\boldsymbol{\kappa}}(\boldsymbol{c}^{i-1}), \tag{6}$$

$$\hat{s}^i, \boldsymbol{c}^i = f_{\boldsymbol{\theta}}(\boldsymbol{y}^i, \boldsymbol{c}^{i-1}) = f_{\boldsymbol{\theta}}(\boldsymbol{d}_{\boldsymbol{\phi}^i} \odot \boldsymbol{x}^i, \boldsymbol{c}^{i-1}). \tag{7}$$

We hypothesize that it would be beneficial to add memory to the context vector, so that it does not contain only information about the previous state of task model $f_\theta(.)$ but can contain information from even earlier acquisition steps. To this end, we propose to leverage a recurrent neural network structure in the form of a Long Short-Term Memory (LSTM) cell (Hochreiter & Schmidhuber, 1997). This LSTM is integrated into the neural networks as a separate layer. We visualize the architecture of the A-DPS framework in Fig.1 and discuss its computational complexity of A-DPS in Appendix A.

## 4 EXPERIMENTS

To show the applicability of A-DPS on both classification as well as reconstruction tasks we evaluate its performance in two experiments. First, we will compare A-DPS with DPS at different subsampling ratios on an MNIST classification example in Section 4.1. Second, we will compare A-DPS with contemporary CS and deep learning methods on a MRI example in Section 4.2, leveraging the fast MRI knee dataset (Zbontar et al., 2018).

### 4.1 MNIST

**Experiment setup**  Classification performance at different sampling rates was tested on the MNIST database (LeCun et al., 1998), consisting of 70,000 grayscale images of $28 \times 28$ pixels of handwritten digits between 0 and 9. We keep the original data split of 60,000 training and 10,000 testing examples. We train both DPS top-M and A-DPS to take partial measurements in the pixel-domain at different sampling rates.

Reaching back to Fig.1 and equations (6) and (7), DPS top-M sampling only consists of the sampling and task model ($f_{\boldsymbol{\theta}}(.)$) parts. All $M$ samples are selected at the same time and used once by $f_{\boldsymbol{\theta}}(.)$ to predict which digit the network is looking at. In the case of A-DPS however, only 1 sample is taken at a time and used as input for $f_{\boldsymbol{\theta}}(.)$. Here, $f_{\boldsymbol{\theta}}(.)$ also creates a context that is used by the sampling network $g_{\boldsymbol{\kappa}}(.)$ to select the next sample. A-DPS iterates through this loop $M$ times in order to select all the samples. We keep $f_{\boldsymbol{\theta}}(.)$ the same for both DPS and A-DPS. Resulting in the fact that the last iteration of A-DPS is similar to that of DPS top-M (i.e. $M$ samples are selected and fed through $f_{\boldsymbol{\theta}}(.)$).

**Task model**  In the classification network $f_{\boldsymbol{\theta}}(.)$, all 784 ($28 \times 28$) zero-masked samples are used as input for 4 fully connected layers. The fully connected layers have 784, 256, 128, and 128 nodes, respectively. Moreover, they are all activated by leaky ReLU activation functions with a negative slope of 0.2. The first three layers also have a dropout of 30%.

The output of the last fully connected layer is then fed into an LSTM with a hidden size of 128. The output of the LSTM is both used as the context for the sampling network, as well as the input for one final fully connected layer. This final fully connected layer maps the output of the LSTM into the 10 class labels, and is therefore followed by a softmax activation function.

In case of A-DPS the output of the LSTM is also used as the input for the sampling network $g_{\boldsymbol{\kappa}}(.)$. The sampling network consists of two linear layers with output sizes of 256 and 784, respectively. Moreover, after the first layer a leaky ReLU activation function is used with a negative slope of 0.2, and a dropout of 30% is applied.

**Training details**  Both sampling strategies were trained to minimize the categorical cross-entropy loss. The temperature parameter was fixed to 2. We again employ SGD with the Adam solver (lr $= 2e - 4$, $\beta_1 = 0.9$, $\beta_2 = 0.999$, and $\epsilon = 1e - 7$) to minimize the loss function. Training was performed on batches of 256 examples for 50 epochs.

**Results** The resulting accuracy on the test set is shown in Fig. 2a. A-DPS outperforms DPS especially when the actual number of samples taken is very low. It is hypothesized that it is especially important to select those candidate samples that carry a lot of information based on the previous samples for very low data rates. Two examples of the selected sampling masks at $r = 4\%$ are displayed in Fig. 2b. Here, it is shown how DPS selects all samples at once, while A-DPS selects them in an iterative fashion, resulting in different sampling patterns for the two examples.

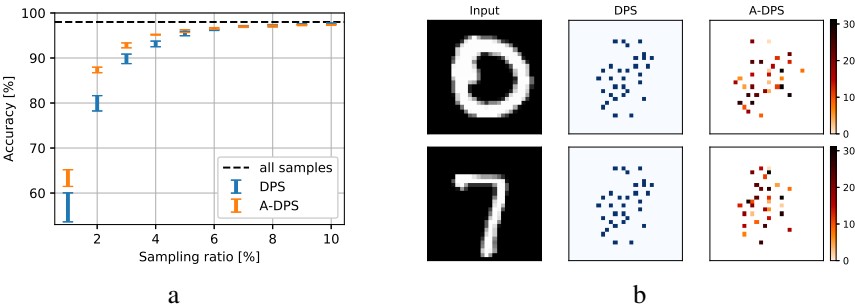

a                                b

Figure 2: Results for the MNIST experiment. a) Classification accuracy on the test set for different sampling ratios. Confidence bounds of one standard deviation are shown. The 'all samples' line indicates the accuracy achieved without subsampling. b) Comparison of the sampling masks for two examples at $r = 4\%$. The colors denote (for A-DPS) the order in which the samples were taken.

To study the effect of the LSTM in this set-up, we perform an ablation study by replacing it with a fully connected layer, resulting in a direct projection of the context vector from the previous state without any further memory. The results of this ablation study are shown in Fig. 3a. Here we can see that for the stricter sampling regimes below 3% the use of an LSTM leads to a higher accuracy.

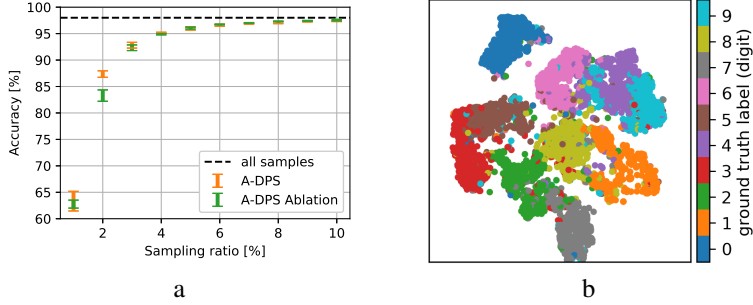

a                                b

Figure 3: a) Classification accuracy on the test set for different sampling ratios for A-DPS and an ablated version of A-DPS where the LSTM has been replaced by a fully connected layer. Confidence bounds of one standard deviation over 5 runs are shown. b) A t-SNE plot of the sampling patterns - colored by digit - shows that A-DPS actively generates different sampling masks for different digits

To analyze the sampling patterns created by A-DPS we create a t-SNE plot (Van Der Maaten & Hinton, 2008), which maps the created sampling pattern (for $r = 4\%$) per image in the test set to one dot in a 2D space. In this 2D space t-SNE aims to preserve spatial relationships from the higher dimension, i.e. similar high dimensional vectors get mapped close to together, while dissimilar ones are mapped further apart. The resulting plot is shown in Fig. 3b, where each dot is colored with the ground truth label (digit) of the corresponding image. The clustering in this figure indicates how similar selected sampling patterns are. For example, the sampling patterns for digit zero tend to be dissimilar from all the other sampling patterns, while the digits four, six, and nine are sampled more similarly using A-DPS.

To get more insight into the inner workings of A-DPS, we perform a similar t-SNE analysis on the context vector that A-DPS constantly updates after acquisition of a new sample. The result is shown in Fig. 4. Here we show the same scatter plot with two different colorings, namely the current acquisition step and the ground truth label (digit). It can be seen how the context vectors are similar

for the first few acquisition step, but as information comes in the context vectors accumulate useful information and branch of into different regions dependent on the ground truth label.

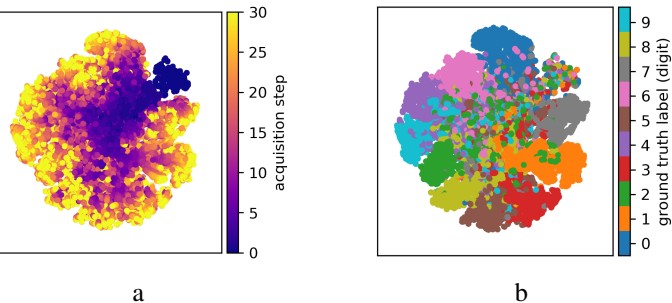

a                                         b

Figure 4: A t-SNE plot of the generated context vector by A-DPS, for all images in the test set. This context vector changes every time a new sample is acquired. This change is visualized by labeling each dot with the number of samples acquired so far (a), showing that context vectors become more distinct after acquiring more data. (b): The same plot, but now colored by the ground truth label (digit).

## 4.2 MRI

**Experiment setup**  To show the applicability of A-DPS, we demonstrate its performance on line-based MRI. We make use of the NYU fastMRI database of knee MRI volumes (Zbontar et al., 2018). Only the single-coil measurements were selected, from which the outer slices were removed. The resulting data was split into 8,000 training, 2,000 validation, and 3,000 testing MRI slices. All slices were cropped to the central $208 \times 208$ pixels and normalized between 0 and 1. The subsampling operation on one of these MRI slices is then performed in k-space (Fourier-space):

$$\boldsymbol{Y} = |\mathcal{F}^H \boldsymbol{D} \odot \mathcal{F} \boldsymbol{X}|, \tag{8}$$

where $|.|$ is the magnitude operator. Moreover, $\boldsymbol{X} \in \mathbb{R}^{N \times N}$ is the fully sampled ground truth image and $\boldsymbol{Y} \in \mathbb{R}^{N \times N}$ is the subsampled image, both in the pixel domain. In this case $N$ is equal to 208. Furthermore, $\mathcal{F}$ and $\mathcal{F}^H$ denote the forward and inverse 2D-Fourier transform, respectively. $\boldsymbol{D} \in \{0, 1\}^{N \times N}$ denotes the sampling mask in k-space.

Normally $\boldsymbol{Y}$ would be complex, due to the asymmetrical nature of MRI measurements and the incomplete subsampling mask. Here, we choose to take the magnitude of $Y$ to simplify reconstruction. We hypothesize that doing so does not significantly change the problem, as the imaginary part of fully sampled images in the NYU fastMRI dataset is very small compared to the real part.

**Task model**  To reconstruct an estimate of the original image $\hat{\boldsymbol{X}}$ from the partial measurement $\boldsymbol{Y}$ a deep unfolded proximal gradient method is used (Mardani et al., 2018), in which $K$ iterations of a proximal gradient method are unfolded as a feed forward neural network following:

$$\hat{\boldsymbol{X}}^{(k+1)} = \mathcal{P}^{(k)}_{(\zeta)} \left\{ \hat{\boldsymbol{X}}^{(k)} - \alpha^{(k)}_{(\psi)} \left( |\mathcal{F}^H \boldsymbol{D} \odot \mathcal{F} \hat{\boldsymbol{X}}^{(k)}| - \boldsymbol{Y} \right) \right\}, \tag{9}$$

where $\mathcal{P}^{(k)}_{(\zeta)}(.)$ is a trainable image-to-image proximal mapping and $\alpha^{(k)}_{(\psi)}$ is the step size, parameterized by $\zeta$ and $\psi$, respectively. We implement this proximal gradient method for $k = 3$ steps, with the trainable step size $\alpha^{(k)}_{(\psi)}$ implemented as a $3 \times 3$ convolutional layer. Each proximal mapping is implemented as a series of 4 convolutions with 16, 16, 16, and 1 feature(s) each and a kernel size of $3 \times 3$. All convolutions but the last are followed by ReLU activation functions.

We will compare A-DPS to to several relevant sampling baselines, namely, random uniform, low-pass, variable density (VDS), greedy mask selection (Sanchez et al., 2020), LOUPE (Bahadir et al., 2019; Bahadir et al., 2020), and DPS. Under a random uniform regime all $N$ lines are equally likely to be sampled, while under a low-pass regime the $M$ lines closest to the DC frequency will be selected. VDS on the other hand is a heuristic regime that employs a probability density from which

the desired amount of samples are drawn. Following (Lustig et al., 2007), we here use a polynomial probability density function with a decay factor of 6.

For the greedy mask selection we follow the approach by Sanchez et al. (2020) and first optimize the sampling mask using the NESTA solver (Becker et al., 2011). After this, we fix the sampling mask and train our proximal gradient network. Results for both reconstruction algorithms are reported.

To generate a sampling mask $D$ using A-DPS we use a sampling network $g_\kappa(.)$. This network takes two inputs. Firstly, the output of the proximal network, i.e. the reconstructed image at that iteration. This image is analyzed using 3 convolutional layers with kernels sizes of $3 \times 3$ followed by ReLU activation functions. The output features are of sizes 16, 16, and 32, respectively. The final feature map is aggregated into a feature vector using global average pooling. Next to this feature vector, the indices of the selected lines at the previous iteration(s) are also used as input for the sampling network, encoded as a $M$-hot vector of dimension 208. Both the feature and selected indices vector are concatenated and used as input for an LSTM cell, with output dimension of 208. This is followed by two fully connected layers with 208 neurons each, having a ReLU activation after the first layer.

**Training details** To promote the reconstruction of visually plausible images, we leverage both a Mean Squared Error (MSE) and adversarial loss (Ledig et al., 2016). To that end we introduce a discriminator network that is trained to distinguish between real and reconstructed MR images. The discriminator is implemented using three convolutional layers with kernel sizes of $3 \times 3$, stride 2, and 64 feature maps, each with Leaky ReLU activations. After the last convolutional layer the feature maps are aggregated into a feature vector using global average pooling, with a dropout rate of $40\%$, which is mapped to a single output probability using one fully connected layer followed by a sigmoid activation function. Next to the MSE loss and adversarial loss, we add a third loss term that penalizes the MSE loss between the discriminator features of real and generated images. The total loss function is a weighted summation of these three losses, with weights 1, $5e-6$, and $1e-7$, respectively.

All sampling mask selection strategies were then trained using SGD on batches of 4 images for a total of 10 epochs. We again employ the Adam solver (lr = $2e-4$, $\beta_1 = 0.9$, $\beta_2 = 0.999$, and $\epsilon = 1e-7$) to minimize the loss function, and set the temperature parameter to 2. We choose $M = 26$, which results in a sampling ratio of $r = 12.5\%$, or an acceleration factor of 8.

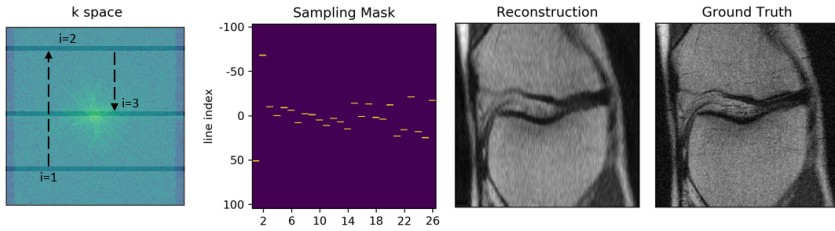

Figure 5: A-DPS MR image reconstruction of a test-set image by adaptively sampling 26 k-space lines. From left to right: 1) k-space with the first 3 sampled lines, 2) sequence of line indices selected by A-DPS, 3) reconstructed image, 4) fully sampled MR image (ground truth).

**Results** We score the different strategies based on 3 metrics: the normalized mean square error (NMSE), the peak signal-to-noise ratio (PSNR), and the structural similarity index (SSIM) (Wang et al., 2004). Averaged results over 5 runs for an acceleration factor of 8 are shown in Table 1, while those for an acceleration factor of 8 are shown in Table 2. We test for the statistical significance of the gains made by A-DPS over DPS in Appendix B. An example of A-DPS reconstruction is shown in Fig. 5, while a comprehensive overview of all baselines for this example can be found in Appendix C and D.

To analyse the workings of A-DPS in this MRI setting we first plot the relative occurrence of different line indices over the test set. This is shown in Fig. 6a. We can see that A-DPS always selects a band of lines around 0, with a lower occurence of high frequency lines. We also employ t-SNE on the context vector that A-DPS updates every acquisition step. The result of this is shown in Fig. 6b. It can be seen that untill acquisition step 15 the context vectors are very similar for the images per

acquisition step, while after acquisition step 15 the context vectors start fanning out. It is hypothesized that at the start of the sampling procedure not a lot of actionable information is available to the system, but this increases as more samples are taken over time.

Table 1: Averaged results over five runs on the hold-out test set for an acceleration factor of 8.

| Sampling Model | Reference | NMSE | PSNR | SSIM |
|---|---|---|---|---|
| Random uniform | | 0.324 | 16.0 | 0.47 |
| Low pass | | 0.043 | 25.2 | 0.71 |
| Variable density | (Lustig et al., 2007) | 0.041 | 25.4 | 0.73 |
| Greedy Mask NESTA | (Sanchez et al., 2020) | 0.043 | 24.8 | 0.71 |
| Greedy Mask Prox. Grad. | | 0.041 | 25.4 | 0.73 |
| LOUPE | (Bahadir et al., 2019) | 0.045 | 25.7 | 0.74 |
| DPS | (Huijben et al., 2020a) | 0.039 | 25.7 | 0.74 |
| A-DPS (proposed) | | **0.036** | **26.1** | **0.75** |

Table 2: Averaged results over five runs on the hold-out test set for an acceleration factor of 16.

| Sampling Model | Reference | NMSE | PSNR | SSIM |
|---|---|---|---|---|
| Random uniform | | 0.265 | 18.9 | 0.5 |
| Low pass | | 0.060 | 23.4 | 0.65 |
| Variable density | (Lustig et al., 2007) | 0.058 | 23.4 | 0.66 |
| Greedy Mask NESTA | (Sanchez et al., 2020) | 0.068 | 22.3 | 0.63 |
| Greedy Mask Prox. Grad. | | 0.059 | 23.4 | 0.66 |
| LOUPE | (Bahadir et al., 2019) | 0.206 | 20.2 | 0.55 |
| DPS | (Huijben et al., 2020a) | 0.056 | 23.8 | 0.67 |
| A-DPS (proposed) | | **0.049** | **24.4** | **0.69** |

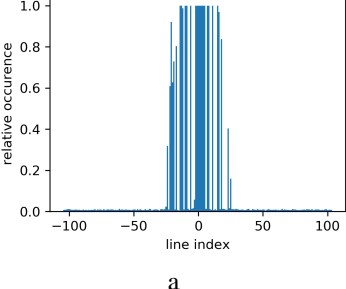
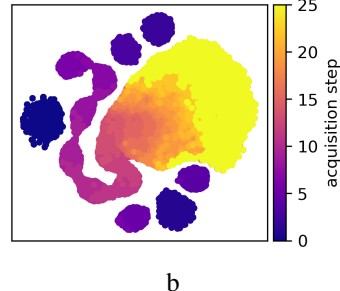

a
b

Figure 6: Analysis of A-DPS for MRI. a) Relative occurrence of the line indices. b) t-SNE analysis on the context vector that is updated every acquisition step. The coloring corresponds to the current acquisition step, which is equal to the number of sampled k-space lines.

## 5 CONCLUSION

We proposed a generalization of DPS, which enables active acquisition, called A-DPS. We demonstrated its applicability on both an MNIST classification task as well as an MRI reconstruction task. Moreover, we found that the adaptive nature of A-DPS improves performance over other sampling pattern selection methods on downstream task performance. We find that A-DPS uses qualitatively differing sampling strategies depending on the context. On a critical note, the black-box nature of A-DPS comes with the traditional machine learning challenges of out-of-distribution generalization and overfitting. This means that in a practical application, the sub-sampling regime could obfuscate the information required to recognize failure cases. Future work includes exploration on how to improve conditioning of the sampling scheme on earlier acquired information and meta-information (such as resolution, sampling ratio, and weighting). Potential future applications include 3D and dynamic MRI, CT, ultrasound, radar, video, and MIMO systems.

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

## A   COMPUTATIONAL COMPLEXITY

One of the drawbacks of A-DPS compared to learned fixed sampling schemes is it higher amount of computational complexity. The main source of this complexity is the unrolling of iterations, leading to a computational complexity of $O(I) = O(M/\rho)$. Although we set $\rho$ equal to 1 in all our experiments, one can in fact seamlessly interpolate between A-DPS and DPS by choosing $1 \leq \rho \leq M$. This constitutes a trade-off between computational complexity and adaptation rate. We leave further exploration of this trade-off to future work.

We can also express computational complexity in terms of run-time on a machine, in our case a GeForce GTX 1080 Ti. A comparison of DPS and A-DPS in terms of training time per epoch can be seen in Fig. 7. We can see that the training time for A-DPS increases for higher sampling ratios where it needs to unroll through more iterations. By combining the results from Fig. 2a and Fig. 7 one can make a trade-off between run-time and accuracy. Where A-DPS achieves higher accuracy for stricter sampling regimes, while at the same time not increasing run-times by a lot.

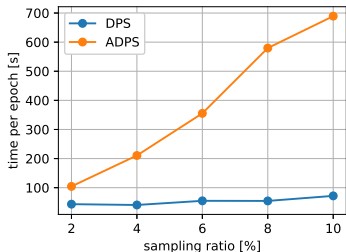

Figure 7: Comparison between DPS and A-DPS of time taken to train for one epoch for the MNIST example on a GeForce GTX 1080 Ti.

For the MRI experiment the training times per epoch are 2 and 52 minutes, for DPS and A-DPS, respectively. Inference is however fast: A-DPS only requires $\sim 13\,ms$ of processing time to determine the next-to-acquire K-space line and reconstruct an image after each step. This is well below the shortest reported Time of Echo (TE) for this MRI acquisition, being 27 ms (Zbontar et al., 2018).

# B    STATISTICAL TESTS ON MRI

To analyse whether the gains made by A-DPS over DPS are statistically significant we perform two-sided paired Student's t-tests, the results of which are shown in Table 3 and Table 4. We perform two different types of tests. Firstly, we look at the average results for each run after training on the hold out test set. This results in a t-test with n = 5 that conveys how reliably A-DPS outperforms DPS, given a new random initialization of the trainable network weights, and different stochastic optimization behavior. These results are shown in Table 3. Secondly, we perform a t-test over the results on each individual image (averaged over the 5 runs) of the hold-out test set, resulting in n = 3000. This test indicates whether A-DPS' performance is significantly higher than that of DPS, given a new test image. These results are shown in Table 4.It can be seen how in all cases for all metrics $p < 0.05$ indicating that our findings are statistically significant.

Table 3: Student's t-test performed over the average results of the 5 runs (n = 5), indicating how reliably A-DPS outperforms DPS given a new random initialization.

|  | 8 times acceleration | | 16 times acceleration | |
|---|---|---|---|---|
|  | P value | T-statistic | P value | T-statistic |
| NMSE | 0.0272 | -2.82 | 0.00652 | -3.65 |
| PSNR | 0.0217 | 2.90 | 0.00798 | 3.57 |
| SSIM | 0.0170 | 3.06 | 0.00852 | 3.56 |

Table 4: Student's t-test performed over the average results on each test image (n = 3000), indicating how reliably A-DPS outperforms DPS given a new image.

|  | 8 times acceleration | | 16 times acceleration | |
|---|---|---|---|---|
|  | P value | T-statistic | P value | T-statistic |
| NMSE | 1.44 e-5 | -4.34 | 3.26 e-12 | -6.99 |
| PSNR | 6.40 e-16 | 8.10 | 2.92 e-41 | 13.6 |
| SSIM | 6.10 e-25 | 10.4 | 5.05 e-18 | 8.70 |

## C MRI RECONSTRUCTION EXAMPLES FOR ACCELERATION FACTOR 8

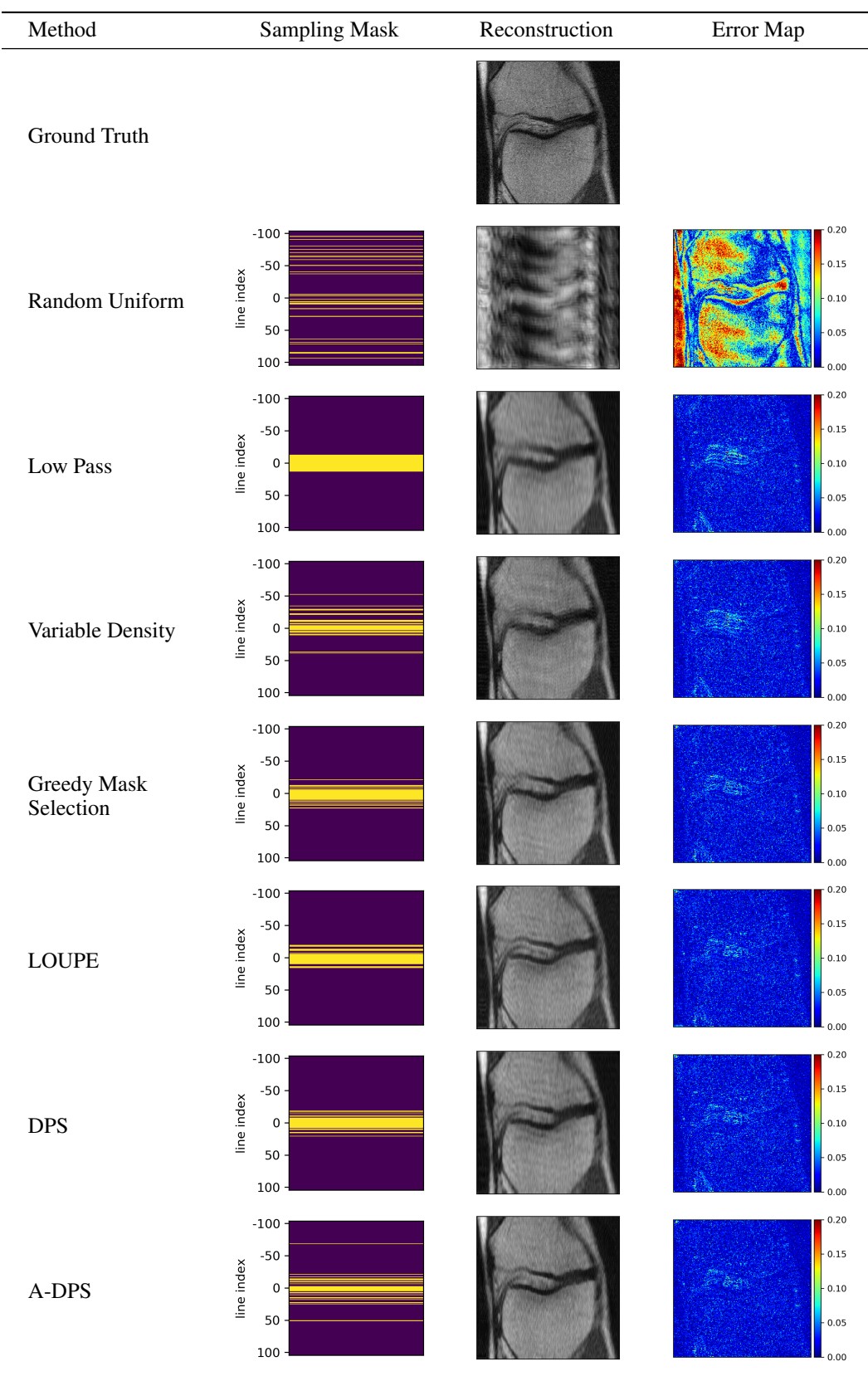

# D  MRI RECONSTRUCTION EXAMPLES FOR ACCELERATION FACTOR 16

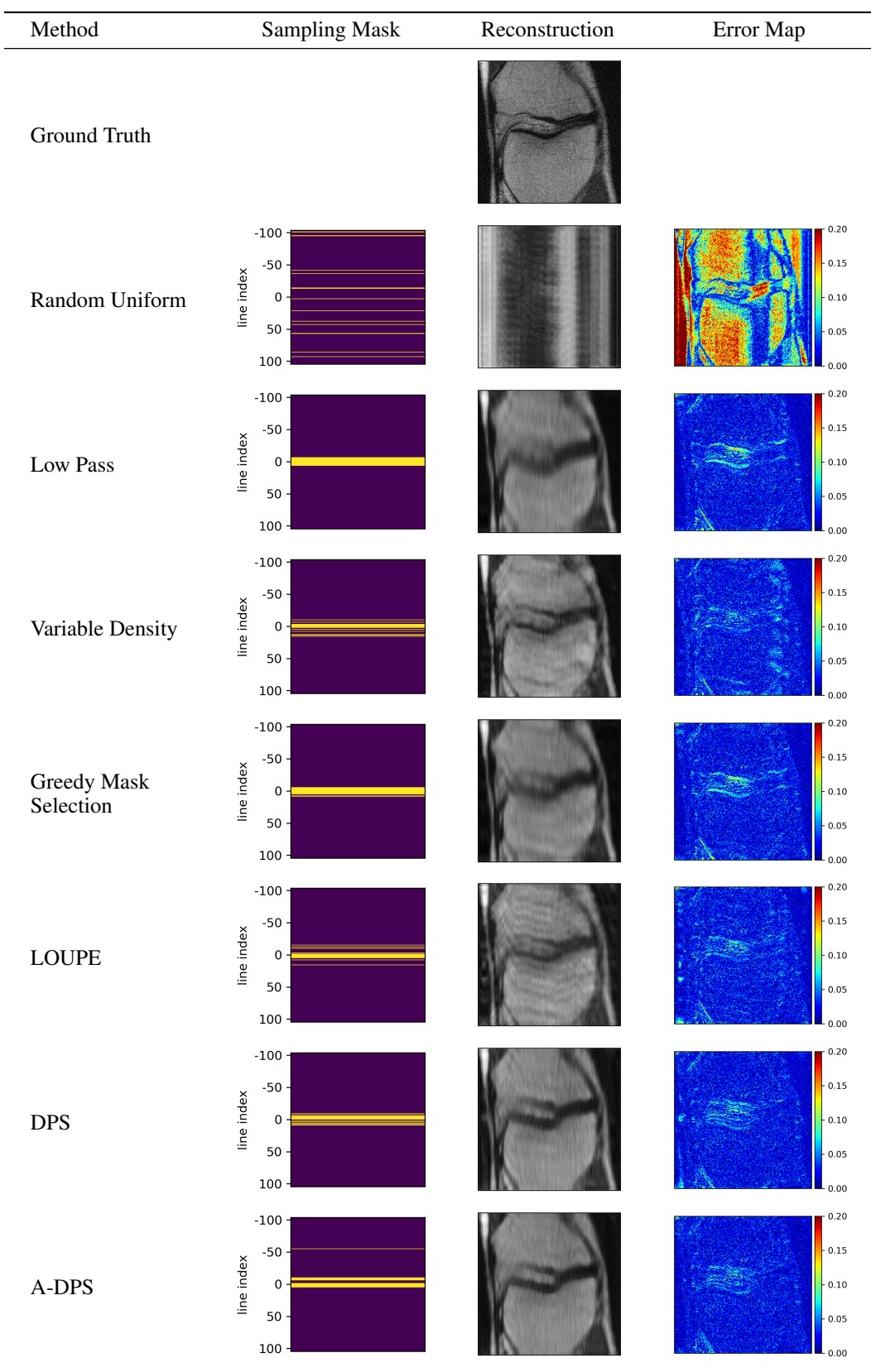

