# OpenReview forum: "Active Deep Probabilistic Subsampling"
_ICLR.cc/2021/Conference — Reject_

### Official Review · AnonReviewer1 · 2020-10-26
**Interesting extension of an existing method with somewhat limited experiments**

**Rating:** 6
**Confidence:** 4

**Review:**

SUMMARY:
The paper at hand deals with compressed sensing (CS) and introduces an extension to deep probabilistic subsampling (DPS) called active deep probabilistic subsampling (A-DPS): instead of learning a sampling pattern that is equal for each element of the dataset, A-DPS adaptively selects entries (of each element) based on the information acquired so far. It is shown that this active sampling increases performance for different tasks: a toy example that aims to demonstrate the benefits of active sampling, a classification task (from subsampled inputs) on the MNIST dataset and a reconstuction task on the NYU fastMRI database of knee MRI volumes.

STRENGTHS:
1. The paper is very clearly written and very comprehensible. Furthermore, it is very detailed about the experimental setup. I also liked the description of the general framework which thoroughly defines the used notation.
2. The idea is well motivated and the approach of selecting samples depending on the previously selected ones makes intuitively sense.
3. The results of the experiments on MNIST and the NYU fastMRI data are promising. A plenthora of (non-active) subsampling schemes are benchmarked as well.

WEAKNESSES:
1. The greatest weakness of this paper is the missing comparison to other active sub-sampling schemes (Zhang et al., 2019; Jin et al., 2019). It would be nice to see wether the proposed method produces better results than the existing methods.
2. I found the toy example very constructed. It is not really easy to understand and does in my opinion not improve the quality of the paper.

QUESTIONS:
- What happens when the MNIST sampling ratio in Figure 3a is further increased? Does A-DPS consistently outperform DPS in low sampling ratio regimes?

DECISION:
Overall, the paper presents an interesting and novel approach. However, it remains an open question wether the proposed A-DPS scheme performs better than already existing active subsampling schemes. Besides this, the experimental evaluation is solid. I lean towards acceptance.

UPDATE AFTER REBUTTAL:
I thank the authors for their responses and appreciate the inclusion of some of the requested changes in the paper. However, the paper still misses the comparison to other adaptive methods which is the paper's greatest weakness. Therefore, I decided to keep my score at 6.

MINOR REMARKS:
- Caption of Table 1 could use some more spacing

---

> ### Author Response · Authors · 2020-11-14
> **Reply to review #1**
>
> We would first like to thank the reviewer for the constructive feedback and suggestions. We will use this to strengthen our paper across this discussion period. In the following we already provide an initial reply to the questions and concerns:
>
> [Weakness 1]:
> We follow the reviewers advice and across the next days we will do our utmost to further enhance the results section through comparison and in-depth model insight.
>
> [Weakness 2]:
> We follow the reviewers advice and will remove the toy example in the revised paper.
>
> [Question 1]:
> We agree with the reviewer that this would be an interesting experiment. We thus evaluated the impact of further increasing the subsampling rate in the MNIST example. Reducing the sampling rate to 1\% leads to average accuracies of 63\% and 57\% for A-DPS and DPS, respectively, confirming the trend that A-DPS consistently outperforms DPS in the low sampling rate regime. Note that this sampling is very sparse (only 7 samples in total) explaining the reduced accuracies. We will update Fig. 3a in the revised manuscript by including these new results.

---

> ### Author Response · Authors · 2020-11-25
> **Second reply to review #1**
>
> We again would like to thank the reviewer for the constructive feedback and suggestions that helped to improve our paper. As a result, we have made the following changes to the manuscript:
>
> [Weakness 1]: We would like to refer the reviewer to the above general response about comparison to other active acquisition baselines. We fully agree that now that several code releases (concurrent to this submission) have become available, future work should include such a comparison.
>
> [Weakness 2]: Following the advice of the reviewer we have removed the toy example from the manuscript as it was deemed confusing and distracting from the main message of the paper.
>
> [Question 1]: We have updated Figure 3a (Now Figure 2a, as the toy example was removed) with the additional experiment that we performed at a 1% sampling ratio.

---

### Official Review · AnonReviewer3 · 2020-10-28
**A borderline case?**

**Rating:** 6
**Confidence:** 3

**Review:**

In this paper, the authors consider the problem of compressed sensing where the underlying signal of interest is captured and restored based only on sparse measurements: Specifically, this paper focuses on the scenario of Deep Probabilistic Subsampling (DPS) which finds sparse measurements in the way that the models designed to solve specific learning problems based on these measurements are jointly optimized. The authors extend DPS to a sequential framework that iteratively and actively selects the next measurement points: The proposed approach encodes the information accumulated until a time step into a context vector which is updated, and used in selecting the next point, in an LSTM-like framework (see minor comments below). In the experiments with two toy problems (including MNIST) and an MRI reconstruction problem, the authors demonstrated that the proposed Active DPS (ADPS) outperforms DPS (in toy problems) and three other compressed sensing algorithms (for MRI reconstruction).

I think this paper makes a borderline case: DPS provides a framework that combines the compressed sensing part (sparse data acquisition) and the subsequent learning part in an end-to-end manner. This paper contributes by extending DPS into an active/sequential learning framework achieving significant performance gains over DPS (mainly on toy problems. see minor comments below). On the other hand, the proposed approach appears to be incremental: ADPS adds a simple sequential update structure (of a context vector) to DPS, which can be described by only two equations (6 and 7). The simplicity of the changes proposed (over DPS) is not a limitation, but it could be accompanied by an in-depth theoretical analysis, a convincing qualitative discussion or _extensive_ experiments demonstrating the practical relevance of the proposed approach.

Minor comments
- Apart from the last one paragraph, the Introduction Section focuses on discussing the context and motivation of Deep Probabilistic Subsampling (DPS). Instead, the authors could use this space to describe and characterize the proposed Active DPS in detail.
- I was not sure why the proposed architecture (Figure 1 and equations 6 and 7) is called LSTM, it has a recurrent network structure but I was not able to find any attention (gating) mechanism that characterizes LSTM. Please advise me if I missed anything.
- Please test if the improvements gained by ADPS over DPS on MRI reconstruction are statistically significant.

Update:

Thank the authors for their responses, clarification, and additional experiments. I read through authors’ responses and the comments from the other reviewers. I still think this paper makes a borderline case for 1) its technical contribution on extending DPS and thereby achieving significant performance gain on a toy problem and MRI reconstruction tasks, still 2) with limited novelty and room for a more extensive experimental validation (perhaps, beyond MRI). My other concerns on clarity and significance of experiments have been addressed. I would raise my rating to marginally above acceptance threshold (borderline).

---

> ### Author Response · Authors · 2020-11-14
> **Reply to review #3**
>
> We would first like to thank the reviewer for the constructive feedback and suggestions. We will use this to strengthen our paper across this discussion period. In the following we already provide an initial reply to the questions and raised concerns:
>
> [Main review]:
> We agree with the reviewer that the paper would improve by providing more substantiation for our method. We follow the reviewers advice and will do our utmost to further enhance the results section through comparison and in-depth model insight.
>
> [Comment 1]:
> We agree with the reviewer that the introduction section focuses a lot on the context and motivation of non-adaptive sampling. We therefore will update the original introduction and related work sections in order to discuss the need for, and advantages of adaptive acquisition methods in more detail.
>
> [Comment 2]:
> To properly answer this question, we must clarify that $f_\theta(.)$ is a deep neural network which consists (among others) of an LSTM cell. This is opposed to the understanding of the referee that our entire model behaves as an LSTM cell. Our framework indeed encapsulates recurrency, but it's not an LSTM cell in itself. We hope this clarifies the raised question. To clarify this concept in the revised manuscript, we will change the text surrounding equations (6) and (7). Explaining in more detail the nature of $f_\theta(.)$ and $g_\kappa(.)$ and how the LSTM fits into them.
>
> [Comment 3]:
> We agree with the reviewer that it is indeed prudent to test whether the improvements gained by A-DPS over DPS on MRI reconstruction are statistically significant, and we will update the revised manuscript with such a statistical comparison.

---

> ### Author Response · Authors · 2020-11-25
> **Second reply to review #3**
>
> We again would like to thank the reviewer for the constructive feedback and suggestions that they have given us. As a result, we have made the following changes to the manuscript:
>
> [Main review]:
> We have added additional experiments to our manuscript, at a 1\% sampling ratio for the MNIST classification task, and at a 16 times acceleration factor for the MRI reconstruction task. Moreover, we have performed an insightful t-SNE analysis that (in addition to the performed quantitative assessment) enables qualitative interpretation of our method. We did this for both the MNIST classification and MRI reconstruction tasks. This analysis and associated discussion is included in the revised paper.
>
> [Comment 1]:
> We have revised the introduction section. Reducing the amount of text devoted to DPS in favor of more substantiation for A-DPS.
>
> [Comment 2]:
> In section 3.3 we have added an additional paragraph clarifying how the LSTM cell fits into our method.
>
> [Comment 3]:
> We have performed a statistical analysis on the performance gains made by A-DPS over DPS in the MRI reconstruction task, concluding that they are indeed statistically significant.
>
> We hope that our changes will be well received and look forward to your final decision on our manuscript.

---

### Official Review · AnonReviewer2 · 2020-10-28
**Limited novelty and validation**

**Rating:** 6
**Confidence:** 4

**Review:**

### Summary
This paper develops methods to perform active subsampling. That is, given some downstream task like classification or image reconstruction, it sequentially selects which elements of an image or signal to sample so as to perform said task. It does so by extending the Deep Probabilistic Subsampling (DPS) method developed by Huijben et al. The proposed method is applied to two problems as well as a simplified, low-resolution MRI reconstruction problem.

### Strengths
Motivation: Active sampling is an interesting idea that has been around for some time, but was often computationally impractical. Thanks to GPUs and deep learning, active sampling is becoming more practical and its interesting to see new work in this direction.

### Weaknesses
Novelty: The method is a small extension to the DPS method where the network that selects which rows to samples is conditioned on the existing measurements.

Validation: The paper did not compare to any other active sampling strategies. The authors made no effort to replicate existing methods.

Clarity:
The Markov chain example in section 4.1 was hard to follow and more distracting than informative. The phrase "the task model gets to sample only one position out of every three" reads as if the model is sampling one position out of every three in the sequence. It took some time before I realized this meant that at every position in the sequence it was probing one of the three states.

Impact: The results with active sampling were only marginally better than results with a fixed (learned) sampling strategy.

Limitations: The method is applicable only to true subsampling problems, not general sensing. That is, one isn't designing the rows of a measurement matrix on the fly but rather selecting which row from an existing matrix (identity in most of the examples) that one would like to sample from.


### Recommendation
The paper's presentation could be improved and it is sorely missing comparisons to other active sampling methods. I don't think the papers novelty is enough to overcome these issues and so I do not believe it is ready for publication.

### Comments
While the proposed method was computationally impractical, active sampling was discussed extensively in [A] from a information theoretical perspective.
[A] Ji, S., Xue, Y., & Carin, L. (2008). Bayesian compressive sensing. IEEE Transactions on signal processing, 56(6), 2346-2356.

Because of the nonlinearity in the forward model, equation (9) is not actually proximal gradient descent. I believe there's a sign(F^HD\circFX) term missing from the (sub) gradient.

### Update

I thank the authors for their comprehensive response. While its unfortunate they couldn't compare to any other active methods, the related work and overall clarity of the paper is significantly improved. The t-SNE plots were informative and interesting. While I have reservations about the paper's lack of comparisons, I think its publication still might be a net positive for the research community.

I have updated my score.

##### Other comments
Let A(X)=F^H D\circ F X. The expression A^H(Ax-Y\circ sign(A(x))) is a subgradient of 1/2|| Y - |A(X)|||^2 but A^H(|Ax|-Y) is not. I would avoid calling (9) projected gradient descent as the "gradient" isn't really a gradient.

"We have performed a statistical analysis on the performance gains made by A-DPS over DPS in the MRI reconstruction task, concluding that they are indeed statistically significant." It would be nice to see confidence intervals in Tables 1 and 2.

#### Questions/comments that do not effect the review:
Why use an LSTM/any network with memory? It seems the next sample depends on the previous samples, but not their order. The ablation study on pg 6 shows that memory helps (at low sampling rates), but I don't understand the intuition why. Could the LSTM just have more capacity?

Typos:
pg 2: "cells.During" space
pg 3: "However, This" capitalization

---

> ### Author Response · Authors · 2020-11-14
> **Reply to review #2**
>
> We would first like to thank the reviewer for the constructive feedback and suggestions. We will use this to strengthen our paper across this discussion period. In the following we already provide an initial reply to the questions and raised concerns:
>
> [Novelty]:
> Active sampling is receiving increasing attention in the research community [1-3], illustrating the non-trivial nature of the problem. The extension from DPS to A-DPS is indeed not a large methodological leap, but provides an effective active acquisition framework nonetheless, and on established ground. Adaptivity brings us one step closer to the theoretical optimum of sub-sampling, and it is certainly worth studying the effect of this improvement in isolation. And although there is value in completely novel frameworks too, an isolated adjustment provides greater clarity in the relative impact of the improvement. Moreover, this leads to a simple method that could facilitate straightforward adoption.
>
> To better emphasize this, we will update the manuscript by adding a more extensive discussion in the introduction explaining the use cases for which such an active acquisition framework is desirable over static acquisition as learned by DPS.
>
> [1] Zizhao Zhang, Adriana Romero, Matthew J. Muckley, Pascal Vincent, Lin Yang, and Michal
> Drozdzal. Reducing Uncertainty in Undersampled MRI Reconstruction with Active Acquisition. 2019.
>
> [2] Kyong Hwan Jin, Michael Unser, and Kwang Moo Yi. Self-Supervised Deep Active Accelerated
> MRI. 2019.
>
> [3] Tim Bakker, Herke van Hoof, Max Welling. Experimental design for MRI by greedy policy search. 2020.
>
> [Validation]:
> We follow the reviewers advice and will do our utmost to further enhance the results section through comparison and in-depth model insight.
>
> [Clarity]:
> We thank the reviewer for this careful assessment and recommendation - we agree that the toy example might distract from the main focus of the paper and will therefore remove it in the revised version.
>
> [Impact]:
> To enable better assessment of the improvement with active sampling we will test the statistical significance of the performance gains in our revised manuscript.
>
> [Limitations]:
> The reviewer is correct: indeed A-DPS selects rows from an existing matrix. This is per design and actually a key strength of A-DPS, allowing for immediate hardware implementations that direct reduce the number of samples taken at the sensing side.
>
> [Comment 1]:
> We thank the reviewer for raising this paper to our awareness, which we will include in our revision.
>
> [Comment 2]:
> We are unsure which nonlinearity the reviewer aims at. The only non-linearity in the forward model is the magnitude operator, which is repeated in equation (9). We would of course be happy to further discuss this across the next days. Thanks!

---

> ### Author Response · Authors · 2020-11-25
> **Seond reply to review #2**
>
> We again would like to thank the reviewer for the constructive feedback and suggestions that helped to improve our manuscript. As a result, we have made the following changes to the manuscript:
>
> [Novelty]: We have updated the introduction section to better reflect the added value of active acquisition over learned, but ultimately, static acquisition. Moreover, we have highlighted how this is an active field of research that is receiving a lot of attention recently by citing two newly published papers in the related work section. We emphasize the novelty and non-triviality of the proposed approach, while highlighting its simple and elegant methodological implementation.
>
> [Validation]:
> We would like to refer the reviewer to the above general response about comparison to other active acquisition baselines. We agree that now that several code releases (concurrent to this submission) have become available, future work should include such a comparison. At the time of submission this was not yet possible however, highlighting the timely nature of this work.
>
> [Clarity]:
> Following the advice of the reviewer we have removed the toy example from the manuscript as it was deemed confusing and distracting from the main message of the paper.
>
> [Impact]:
> We have performed a statistical analysis on the performance gains made by A-DPS over DPS in the MRI reconstruction task, concluding that they are indeed statistically significant.
>
> [Comment 1]:
> We again thank the reviewer for raising this paper to our awareness, it has since been included in both the introduction as well as the related work section.
>
> We hope that our changes will be well received and look forward to your final decision on our manuscript.

---

### Author Response · Authors · 2020-11-25
**Comment regarding comparisons with other active sampling strategies**

We would like to address the reviewers' comment regarding comparisons with other active sampling strategies. In the past two weeks we have done our utmost to also implement such an active sampling baseline.

At the time of submission and the ICLR deadline, unfortunately no public code of other active sampling methods was available. By now, besides ours, 2 other concurrent code repo's have been released: Zhang et al., 2019 together with Pineda et al., 2020 (see https://github.com/facebookresearch/active-mri-acquisition) and Bakker et al., 2020 (see https://github.com/Timsey/pg_mri). The corresponding paper of the latter was even only published on the 30th of October. Despite our efforts across the past 2 weeks, we did not manage to achieve a functional reproduction and comparison with these concurrent releases. We really hope that the reviewer understands - we truly did our best in the given time. Future work should include such baselines, and the fact that these repositories have only become available very recently actually highlights the timeliness of the current paper.

We have however compared A-DPS to a plethora of non-active baselines, all of which are included in our code, which is available from here: https://drive.google.com/file/d/1HF6OtEpzcIPB4UOrS4kR3BQdv1pOsuEh/view?usp=sharing. We hope that such open code sharing facilitates reproducibility and ease of comparison between the implemented non-active baselines and our active acquisition frameworks in the future.

---

### Author Response · Authors · 2020-11-25
**General comment to all reviewers**

We again would like to thank all of the reviewers for their time and the constructive feedback and suggestions that they have given us. We would like to use this official comment to list all of the changes that were made to incorporate the given feedback.

*    We have revised the introduction section. Reducing the amount of text devoted to DPS in favor of more substantiation for A-DPS.
*    We have updated the related work section to include three extra sources. One of which was raised to our attention by reviewer 2 (Ji et al., 2008), and the other two that have been published in this field since the initial deadline of the ICLR 2021 conference (Pineda et al., 2020 and Bakker et al., 2020).
*    In section 3.3 we have further clarified how the LSTM cell fits into our method.
*    Following the advice of the reviewers we have removed the toy example from the manuscript as it was deemed confusing and distracting from the main message of the paper.
*    At the request of Reviewer 1, we have expanded the sampling ratios explored in the MNIST classification task by adding the results at a 1\% sampling ratio.
*    We have performed an insightful t-SNE analysis that (in addition to the performed quantitative assessment) enables qualitative interpretation of our method. We did this for both the MNIST classification and MRI reconstruction tasks. This analysis and associated discussion is included in the revised paper.
*    We enriched our experimental section by performing an additional MRI experiment at a different sampling rate, where A-DPS also outperforms all other baselines.
*    Following the advice of reviewer 3 we have performed statistical tests (i.e. the Student's t-test) to show that the performance gain of A-DPS over DPS in the MRI reconstruction task is statistically significant.

We hope that our changes will be well received and look forward to your final decision on our manuscript.

---

### Decision · Program_Chairs · 2021-01-07
**Final Decision**

**Decision:**

Reject

**Comment:**

The review phase was very constructive, where reviewers raised several opportunities for improvements. The authors did a very good job in their rebuttal, which led some reviewers to change their opinion in a positive direction. Overall, reviewers agree that this is the borderline paper with remaining concerns about the weak experimentation.  The paper was again discussed by the Area Chair and Program chairs.  Due to the competitive nature of the conference and the high bar of experimental evaluations expected by empirical papers, the paper was finally rejected.  We hope authors will use the feedback from the reviews and make a stronger submission in near future.